# Frozen Section of Parotid Gland Tumours: The Head and Neck Pathologist as a Key Member of the Surgical Team

**DOI:** 10.3390/jcm11051249

**Published:** 2022-02-25

**Authors:** Konstantinos Mantsopoulos, Zacharias Bessas, Matti Sievert, Sarina Katrin Müller, Michael Koch, Abbas Agaimy, Heinrich Iro

**Affiliations:** 1Department of Otorhinolaryngology, Head and Neck Surgery, Friedrich-Alexander Universität Erlangen-Nürnberg (FAU), 91054 Erlangen, Germany; zbessas@gmail.com (Z.B.); matti.sievert@uk-erlangen.de (M.S.); mueller.sarinakatrin@gmail.com (S.K.M.); michael.koch@uk-erlangen.de (M.K.); heinrich.iro@uk-erlangen.de (H.I.); 2Institute of Pathology, Friedrich-Alexander Universität Erlangen-Nürnberg (FAU), 91054 Erlangen, Germany; abbas.agaimy@uk-erlangen.de

**Keywords:** frozen section, parotid gland, benign tumour, malignant tumour, head and neck cancer, sensitivity, specificity, accuracy, predictive value

## Abstract

Introduction: The aim of this study was to evaluate the impact of subspecialised head and neck versus general surgical pathologists on the reliability of the histopathologic evaluation during intraoperative consultation. Materials and Methods: The medical records of all patients who underwent a parotidectomy with frozen section between 2006 and 2021 were retrospectively evaluated. The frozen section was evaluated for sensitivity, specificity, accuracy, and predictive value. Assessment by two groups of pathologists (subspecialised head and neck versus general surgical pathologists) was compared, and the nature or types of misdiagnoses compared with final diagnoses on paraffin sections were analysed for the two groups. Results: Our study sample was made up of 669 cases. The mean age of patients was 57.7 years (range: 10–94 years). Of these, 490 patients had a benign lesion (73.2%), whereas 179 patients had a malignant lesion (26.8%). Frozen section had an overall accuracy of 97.6%, sensitivity for malignancy was 91.1%, specificity was 100%, PPV was 100%, and the NPV was 96.8%. The exact histologic subtype in the group of malignant tumours was correctly identified in FS in 89.4% of cases. A comparison of head and neck pathologists versus general surgical pathologists revealed a highly statistically significant difference concerning both overall detection of malignancy (*p* < 0.001) as well as correct identification of the histologic subtype (*p* < 0.001). Conclusion: Involvement of subspecialised head and neck pathologists in the intraoperative consultation for salivary gland tumours results in a gain of 19.8% more sensitivity, underlining the importance of specialisation in salivary gland pathology for the optimisation of frozen section quality.

## 1. Introduction

The diagnostic approach to tumours of the parotid gland is admittedly a staged procedure. In the beginning, the clinical features, in combination with imaging (ultrasound, MRI), serve to categorize the lesions into “surgical” and “non-surgical” entities. Concerning the first group of cases, findings of invasive biopsy modalities (fine needle aspiration cytology, core needle biopsy), in combination with findings on intraoperative consultations (frozen section, FS), aim to aid categorisation of the lesions into benign and malignant [1]. FS represents a final opportunity to verify a preoperative clinical suspicion or an indeterminate finding on cytology, in order to aid proper intraoperative decision-making and to facilitate further surgical treatment [2]. The relevance of FS for parotid gland tumours is further solidified by the fact that a fraction of low-grade salivary gland carcinomas are almost indistinguishable from adenomas, and hence, diagnosis of malignancy represents a surprise diagnosis. This is particularly true for centres that do not rely much on preoperative fine needle aspiration cytology (FNAC) due to several factors. 

Ackerman and Ramierez describe the frozen section as “one of the most important and difficult procedures pathologists perform during their practice, as it requires experience, knowledge of clinical medicine, the ability to make quick decisions under time pressure, good judgment, a conservative but also decisive attitude and a keen awareness of the limitations of the method” [3]. Moreover, deep knowledge concerning the clinical-surgical consequences of pathological diagnoses is one of the most critical capabilities a pathologist should gain, and is a prerequisite for well-thought-of decision-making at the time of FS. This is mainly due to the lack of any consequence-free opportunity to reverse FS diagnoses, as the surgical consequence usually has happened already. Being the literal ‘hot seat’ for the pathologist, FS as a diagnostic modality enjoys a privileged place in the interaction between pathology and clinical medicine. In the context of malignant parotid tumours, the high diversity of clinical entities with highly heterogeneous and overlapping tumour morphology, varying tumour biology, and oncologic behaviour poses additional challenges to the pathologist during this highly stimulating task.

The aim of this study was to investigate the overall experience of our department with the “hot-topic” of FS for parotid gland tumours. Furthermore, this project aimed at evaluating the effect of experience and expertise in the field of salivary gland pathology (comparing head and neck pathologists vs. general surgical pathologists) on the quality and reliability of the histopathologic findings on the basis of a single-centre series of 669 cases within a period of 15 years.

## 2. Materials and Methods

This study was performed at an academic tertiary referral centre specializing in salivary gland diseases (Department of Otorhinolaryngology, Head and Neck Surgery, University of Erlangen–Nuremberg, Erlangen, Germany). For this study, the medical records of all patients who underwent a parotidectomy with FS between 2006 and 2021 were retrospectively evaluated. The frozen section was evaluated for sensitivity, specificity, accuracy, and predictive value. Two groups of pathologists—group A (a single subspecialised head and neck pathologist with experience of 10–15 frozen sections of salivary gland lesions per month and more than 150 peer-reviewed articles on this salivary gland pathology (h-index:49)) and group B (general surgical pathologists with no “head and neck” subspecialisation)—were compared against each other for their performance in the FS of malignant lesions. All cases had no prior histopathological diagnosis at the time of FS assessment (cases with a preoperatively known diagnosis, e.g., by means of core needle biopsy were excluded from our study sample).

Sensitivity for malignancy was defined as the number of malignant tumours correctly identified on FS divided by the whole number of true malignant tumours diagnosed on definitive histology. Specificity was defined as the number of benign tumours correctly identified on FS divided by the whole number of true benign tumours detected on definitive histology. Positive predictive value (PPV) was defined as the number of malignant tumours being correctly identified on FS divided by the whole number of tumours being supposed to be malignant tumours on FS. The negative predictive value was defined as the number of benign tumours being correctly identified on FS divided by the whole number of tumours being identified as benign tumours on FS. Accuracy refers to the overall performance of FS in the distinction between benign and malignant parotid gland tumours, and was defined as the sum of all tumours correctly identified on FS divided by the whole number of our study cases.

Statistical analysis was performed using the x^2^ test with 95% confidence intervals (Cis). The software SPSS version 21 for Windows (SPSS, Inc., Chicago, IL, USA) was used for the analysis. A *p*-value of <0.05 was considered statistically significant. The Institutional Review Board (IRB) of the University Hospital of Erlangen approved this study. One of the main aims of this study was to address the issue of whether intraoperative evaluation of parotid gland specimens by different general surgical pathologists (thereby adhering to the duty plan of the institution) represents an acceptable approach, or if it is necessary to involve (secondarily) a specialised head and neck pathologist in cases with ambiguous, rare, or unusual diagnoses upon initial assessment.

## 3. Results

In total, 669 cases made up our final study sample (363 men, 306 women; male-female ratio 1.19:1). Their mean age was 57.7 years (range: 10–94 years). Of these, 490 patients had benign lesions (73.2%), whereas 179 patients had malignant tumours (26.8%). In the same period of time, 2050 cases with parotid gland tumours were managed in our department. Preoperative suspicion for malignancy on the basis of clinical picture and imaging findings was set in 123/179 (68.7%) of the cases with malignant tumours. Taking out the two cases with an indeterminate finding on FS (0.3%), FS set the correct diagnosis (“benign or malignant”) overall in 653/669 cases, giving an overall accuracy (“diagnostic value”) of 97.6%. The calculated sensitivity of FS for malignancy was 91.1% (163/179). In the 16 cases with discordance in the identification of the malignant nature of entity between frozen section and final diagnosis, almost half were misdiagnosed on the frozen section for pleomorphic adenomas (Table 1).

The calculated specificity of FS was 100% (490/490). The PPV was 100% (163/163), giving a false discovery rate of 0%, and the NPV was 96.8% (490/506). The exact histologic subtype in the group of malignant tumours was identified correctly in FS in 160/179 cases (89.4%). A comparison of the experienced head and neck pathologist with the general surgical pathologists revealed a highly statistically significant difference concerning both overall detection of malignancy at FS (*p* < 0.001), as well as correct identification of the histologic subtype (*p* < 0.001, Table 2). 

### Diagnostic Discrepancies between Frozen Section and Permanent Histology

Overall, 16 carcinomas were named benign at FS (Table 1); however, no benign tumour was named malignant. All of these mislabelled entities were either definitionally low-grade carcinomas (*n* = 12) or represented carcinomas ex pleomorphic adenomas (*n* = 4). Submitted diagnoses by pathologist B (general surgical pathologists) was predominantly “no malignancy”, while a diagnosis of pleomorphic (*n* = 3) and monomorphic (*n* = 2) adenoma was favoured in the remainder. Notably, a diagnosis of “no malignancy” and of benign adenoma (oncocytoma or cystadenolymphoma) was rendered with similar frequency by general surgical pathologists, instead of grade 1 mucoepidermoid carcinoma. The same was true for acinic cell carcinoma.

## 4. Discussion

In cases with older, multimorbid patients and suspicion for a cystadenolymphoma (Warthin’s tumour), or in the event of preoperative suspicion of parotid gland malignancy, a core needle biopsy of the lesion is performed in our department. In the first case, confirmation of the diagnosis of a cystadenolymphoma justifies a wait-and-scan strategy by patients with a high perioperative risk [4]. In the case of malignancy, further treatment is decided by an interdisciplinary tumour board. In these cases, a complete parotidectomy with neck dissection is performed. The same treatment modality is chosen in cases in which primary parotid gland malignancy is detected on an intraoperative frozen section. Frozen section examination for use in surgical pathology was introduced near the turn of the last century [5]. In 1905, Louis B. Wilson initiated the use of this diagnostic modality for rapid intraoperative diagnosis at the Mayo Clinic [6]. Starnes et al. [7] and Lester [8] distinguish between the purpose (“intraoperative consultation”) and the means of achieving it (“frozen section” because of the use of cryostat), aiming at a more accurate description of the nature of this diagnostic procedure. This intraoperative dialogue and exchange of information between the surgeon and the pathologist during a period of no more than 30 minutes aims at guiding as well as refining the steps of further surgical management [9] and completion of the surgical treatment in the same session [7]. As a procedure of information exchange between surgery and pathology teams, FS requires a thoughtful collaboration between the involved parties and works best when both sides are proactive about communication and are perhaps willing to take the place of the opposite side [7]. The pathologist should be able to correctly evaluate the clinical information (prior malignant diagnoses, prior treatments, radiotherapy, biopsies, imaging results) accompanying the specimen, communicate a degree of uncertainty in the FS diagnosis, and weigh the potential consequences of a false negative or false positive result. Undoubtedly, a basic understanding of the treatment algorithms being influenced through the FS diagnosis is of major significance. On the other side, the surgeon has to be aware of the limitations of FS, remain reasonable concerning requirements and expectations from the pathology team, and define the extent of consideration of the FS result in the intraoperative decision-making process. In our eyes, the need for “integration” of a clinically oriented head and neck pathologist with experience in the complex pathology of salivary gland tumours and their biology and in treatment strategies in surgical treatment, as reflected in the title of this article, is mandatory for high-quality FS evaluation and for optimised prompt surgical treatment of salivary gland cancer.

The purpose of FS is to provide a sufficient answer to a specific question that has an immediate impact on the intraoperative care of the patient [8]. In the context of parotid gland tumours, the FS primarily aims to determine dignity and, in some cases, to clarify an indeterminate preoperative fine needle aspiration cytology [10]. For malignant tumours, FS enables stratification between lymphomas and primary parotid gland carcinomas as well as “low-” and “high-grade” subtypes. Reliable FS information on the dignity of the parotid lesion should sustain further surgical measures, such as termination of surgery (for benign tumours) or continuation with removal of the whole parenchyma of the parotid gland (e.g., complete parotidectomy in cases with primary parotid malignant tumours) with elective neck dissection [11,12,13] and resection (as well as reconstruction) of the facial nerve in cases of nerve infiltration through a malignant lesion (other than lymphoma) [14]. Intraoperative evaluation of intraparotid lymph nodes could enable rationalised performance of complete parotidectomy and elective neck dissection [15]. Concerning the facial nerve, FS is used to ensure proximal and distal control of the perineural tumour spread before primary nerve reconstruction. Identification of a tumour at the stylomastoid foramen on FS in cases with, e.g., adenoid cystic carcinoma could lead to extension of parotid surgery in the direction of mastoidectomy for control of the proximal margin of the facial nerve. Furthermore, examination of marginal biopsies by means of the frozen section enables optimal determination of the extent of surgery around a malignant parotid tumour. FS of the neck specimen at level II enables rational indication for completion of the neck treatment in the direction of comprehensive management of levels I–V [16,17]. Last but not least, the correct identification of malignancy on FS decreases the patient’s socioeconomic burden, psychologic discomfort, and operative risks (e.g., iatrogenic facial nerve palsy) associated with revision surgery [11]. 

Overall, non-organ-specific accuracy of frozen section is estimated to be about 97.8% [18] to 98.7% [19]. The head and neck region is among the sites with the highest discordance rates in FS [9], with the salivary gland tumours occupying the most unfavourable places in this anatomic region. In general, the reason for discordance between FS and definitive diagnosis could be categorised in diagnostic misinterpretation, problems in block sampling (i.e., the pathologic lesion was present only in deeper permanent sections taken of the frozen section block), or technical errors (e.g., suboptimal quality of the frozen section slide, such as tissue folding). Diagnostic misinterpretation by salivary gland tumours could be justified through the 20 different malignant subtypes [20] (“an evolving art” [11,21]) and the need for further examinations (immunohistochemistry, molecular biology) to diagnose certain subtypes (e.g., secretory carcinoma [22]). According to the relevant literature, FS for malignant parotid tumours has a specificity of 99% (95% CI: 98–99%) [1] with a lowest score of 90% [23] and highest scores as high as 100% [23,24,25,26]. On the contrary, mean sensitivity is much lower with values around 90% (95 CI 81–94%) [1] with the lowest point at 61.5% [27] and highest scores up to 100% [23,24,28] (Table 3).

These data point to the consistently high reliability of FS to correctly detect a benign parotid lesion, and to the extremely varying (moderate to high) ability of FS to reliably identify the presence of malignancy. 

In light of several literature reports, we can identify working groups in high-volume centres with high scores in both parameters [15,23,24], pointing to the crucial role of specialisation and focused experience of the pathologist in this admittedly demanding histopathologic issue. The gain of 19.8% in sensitivity through the performance of FS by an experienced head and neck pathologist in our study sample (98.2% vs. 78.4%, Table 2) emphasises the importance of specialisation in salivary gland pathology for optimisation of FS quality. A comparison of the histopathologic performance scores of the two pathologist groups reveals a higher difference between both groups in the question concerning the histologic subtype of malignancy, but also in the question of “benign or malignant” (Table 2). By examining the quality of FS in association with the percentile FS performance of the experienced pathologist over the years, we could see that after 2013 the improvement in sensitivity for malignant lesions was timely associated with the experienced pathologist’s performance (Figure 1). Evidently, accumulation of relevant experience and expertise in this demanding topic could be reflected by a higher performance in the detection of fine differences between the 20 different histologic subtypes [20].

Another point worth noting is the extremely varying rate of inconclusive or non-diagnostic FS (“deferral rate”) of salivary gland tumours, which ranges between 0% [23,25], 7% [25], 8% [27], 12% [28], and 15% [29] in the relevant literature. With an average deferral rate of 3.7% [1], FS is significantly better than FNA (8.1–9.5% [34]) in this context. Our rate of 0.3% is significantly lower than the aforementioned average value, which could be attributed to the low proportion of samples gained through intraoperative core needle biopsy (1.1%), and points to the fact that increasing expertise enables reliable evaluation of the specimen already on the FS setting. Furthermore, whereas most relevant studies focus on the discordance rate between FS and final diagnoses, very few reports deal with the FS errors that actually negatively impact intraoperative patient management. In a study of almost 25,000 patients (albeit with all kinds of tissue) from the Mayo Clinic, such errors occurred in only 0.1% of the frozen sections performed [18]. In our study, a malignant tumour was misdiagnosed for a benign lesion on FS in 16/179 cases (8.4%), with a highly statistically significant difference in favour of the head and neck pathologist (21.5% vs. 1.8%, *p* < 0.001, Table 2). In these 16 cases, revision surgery (completion or radical parotidectomy with neck dissection) was performed in 10 cases. In the other six cases, the primary site had already been managed by means of subtotal or complete parotidectomy in primary surgery (two cases), revision was refused by the patient (one case), or not indicated due to multiple co-morbidities and high anaesthesia risk (two cases). In a solitary case, the primary tumour was well localised in an accessory salivary gland without any contact with the parotid gland, and was resected with broad margins, so that no revision surgery was required (Table 1). Almost half (7/16) of these cases were misdiagnosed on FS for pleomorphic adenomas without any signs of malignancy. Definitive histology revealed a carcinoma ex pleomorphic adenoma in four out of these seven cases (G1 in three cases and G3 in one case). In these four cases, there is no real discordance, given the correctness of the adenoma diagnosis and the fact that carcinoma ex pleomorphic adenoma is often a reflection of sampling, which is usually limited in FS due to time constraints. Additionally, the diagnosis of carcinoma ex pleomorphic adenoma requires an assessment of the entire periphery of the lesion to look for signs of invasion [5]. Representative examples of discrepant diagnoses of malignant tumours of the parotid gland in frozen section are given in Figure 2.

The drawback of FS as a primary diagnostic modality for salivary gland lesions is mainly a reflection of the complexity of the morphology of salivary gland tumours and of the significant overlap between different benign and malignant entities. In contrast to many other organs, malignancy in salivary gland carcinomas is not always defined by the degree of atypia, except in the high-grade categories (e.g., salivary duct carcinomas, high-grade myoepithelial carcinomas, and others). On the other hand, >50% of salivary carcinomas are “low-grade” and the diagnosis of malignancy is based solely on recognition of the histologic subtype, irrespective of the bland histology and the absence of invasive growth. This explains the difficulty of general pathologists to identify many of the salivary carcinomas correctly at frozen section. Notably, most of the discordant diagnoses by the general surgical pathologists in our current study relate to their limited ability to recognize histological tumour types in bland-looking neoplasms, such as acinic cell carcinoma and mucoepidermoid carcinoma. This is reflected by the frequent use of the diagnosis “no malignancy” (meaning no high-grade morphology at best) in most cases. The only entity discordantly labelled benign by both the general surgical pathologists and the head and neck pathologist is epithelial myoepithelial carcinoma (EMC). This is mostly inevitable, given the close resemblance of EMC to gland-rich pleomorphic adenoma and, in particular, to basal cell adenoma. On the other hand, for some tumours, an adenoma-carcinoma spectrum is established (e.g., low-grade myoepithelial and basal cell neoplasms), so that only a thorough assessment of the gross specimen and careful histology can facilitate recognition of carcinomas and their distinction from adenomas. Moreover, tumours with prior FNAC may show intravascular deposits indistinguishable from “angioinvasive” malignancies [34]. Knowledge of this phenomenon and of extensive bizarre-looking metaplastic changes (metaplastic or infarcted Warthin tumours [35], necrotizing sialometaplasia, etc.) is mandatory to avoid overdiagnosis of malignancy. Generally speaking, for the experienced head and neck pathologists, pitfalls in diagnosing malignant versus benign salivary gland tumours are not very different during FS versus permanent histology, except for the extent of sampling (which can be time-consuming in FS) and the time factor. This is in sharp contrast to the experience of general pathologists, where it is well known that the false-positive and false-negative rates in the diagnosis of malignant versus benign (and vice versa) for several salivary tumour entities are still very high (up to 50% for many low-grade entities). These observations and findings clearly highlight the need for integrating an experienced head and neck pathologist into the surgical team dealing with salivary gland tumours, and that supervision of salivary gland specimens during FS should best be performed with the involvement of the head and neck pathologist if available onsite.

## Figures and Tables

**Figure 1 jcm-11-01249-f001:**
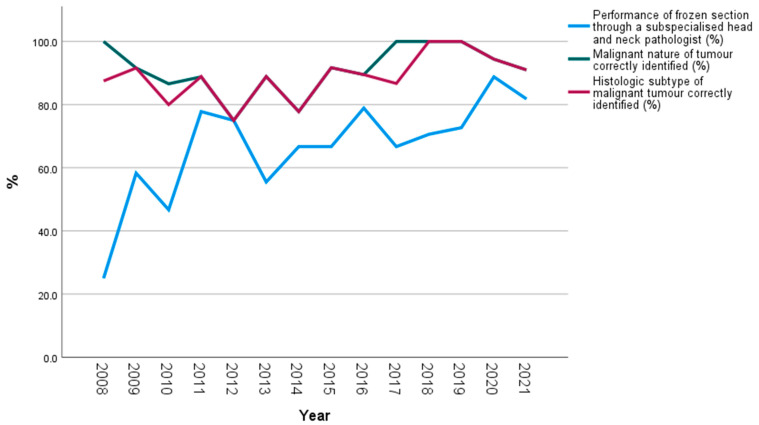
Performance of frozen section by a subspecialised head and neck pathologist, and performance of frozen section concerning correct identification of malignant tumours and their histologic subtypes over the years (the number of examined cases per year is given next to each year in the x axis of the diagram).

**Figure 2 jcm-11-01249-f002:**
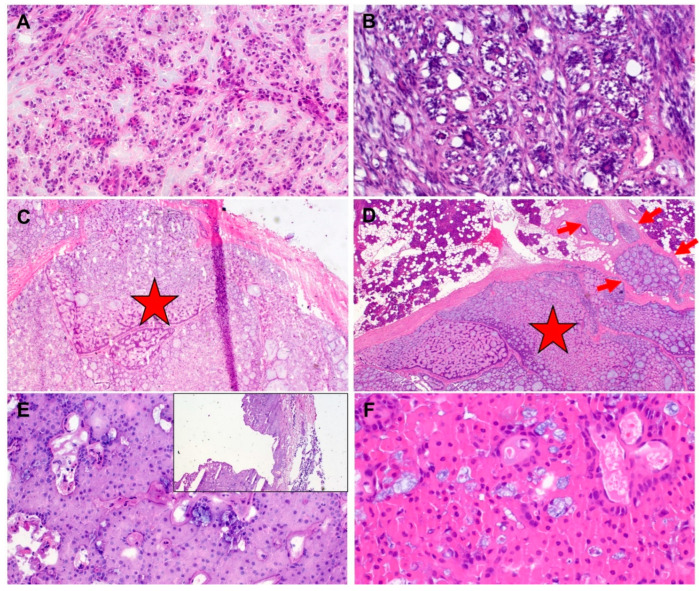
(**A**) (frozen section) and (**B**) (permanent section) of epithelial-myoepithelial carcinoma. This rare entity is essentially not reliably distinguishable from pleomorphic adenoma with diffuse ductal component interrupted by myoepithelial cells (interpretative error). This applies frequently to both general surgical pathologists and head and neck pathologists. (**C**) (frozen section) and (**D**) (permanent section) of adenoid cystic carcinoma with predominantly encapsulated pattern (stars). This interpretative error is frequently seen in cases seen by general surgical pathologists but not by head and neck pathologists. A focal invasive component (red arrows) was also present in the frozen section (interpretative error). (**E**) (frozen section) and (**F**) (permanent section) of cystic and oncocytic mucoepidermoid carcinoma misinterpreted as cystadenolymphoma (“Warthin’s tumor”) by the general surgical pathologist, despite absence of lymphoid tissue and bilayered oncocytes and presence of multifocal mucus cells (interpretative error). Small image: overview showing solid and cystic lesion.

**Table 1 jcm-11-01249-t001:** Detailed demonstration of the study cases with discordance in the identification of the malignant nature of entity between frozen section and final diagnosis (A: experienced head and neck pathologist, B: averagely experienced pathologists).

Case Number	Frozen Section Diagnosis	Definitive Diagnosis	Pathologist	Impact on Management	Source/Type of Error
1	Cystadenolymphoma	G1 mucoepidermoid carcinoma	B	Revision surgery	Interpretative
2	“No malignancy”	Acinic cell carcinoma	B	Revision surgery	Interpretative
3	Pleomorphic adenoma	Basal cell carcinoma ex pleomorphic adenoma	B	No revision surgery (on patient’s request)	Sampling
4	Pleomorphic adenoma	Acinic cell carcinoma	B	Revision surgery	Interpretative
5	“No malignancy”	Acinic cell carcinoma	B	No revision surgery (multiple co-morbidities)	Interpretative
6	“No malignancy”	G1 mucoepidermoid carcinoma	B	Revision surgery	Interpretative
7	Pleomorphic adenoma	Low-grade adenocarcinoma ex pleomorphic adenoma	B	Revision surgery	Sampling
8	Basal cell adenoma	Adenoid cystic carcinoma	B	No revision surgery (complete parotidectomy already performed)	Interpretative
9	Pleomorphic adenoma	Myoepithelial carcinoma ex pleomorphic adenoma	B	Revision surgery	Interpretative
10	Pleomorphic adenoma	Ductal adenocarcinoma ex pleomorphic adenoma	A	Revision surgery	Sampling
11	Pleomorphic adenoma	Epithelial myoepithelial carcinoma	B	Revision surgery	Interpretative
12	Basal cell adenoma	Epithelial myoepithelial carcinoma	A	No revision surgery (subtotal parotidectomy already performed)	Interpretative
13	“No malignancy”	G1 mucoepidermoid carcinoma	B	No revision surgery (multiple co-morbidities)	Interpretative
14	Basal cell adenoma vs. pleomorphic adenoma, “no malignancy”	Secretory carcinoma	B	Tumour in accessory gland, revision surgery not required	Interpretative
15	Pleomorphic adenoma	Adenoid cystic carcinoma	B	Revision surgery	Interpretative
16	Oncocytic adenoma	G1 mucoepidermoid carcinoma	B	Revision surgery	Interpretative

**Table 2 jcm-11-01249-t002:** Comparative performance of highly and averagely experienced pathologists in the histopathologic approach of the malignant cases of our study.

	Performance of Highly Experienced Pathologist (%)	Performance of Pathologists with Average Experience (%)	Total	*p* Value
Malignant nature of entity correctly identified	112/114 (98.2)	51/65 (78.4)	163/179 (91.1)	<0.001
Histologic subtype of malignant tumour correctly identified	112/114 (98.2)	48/65 (73.8)	160/179 (89.4)	<0.001

**Table 3 jcm-11-01249-t003:** Relevant literature reports with data on frozen section of parotid gland tumours (NR: not referred).

Study (Year)	N	Sensitivity (%)	Specificity (%)	Inconclusive or Non-Diagnostic
Carvalho (1999) [27]	153	61.5	98	NR
Tew (1997) [29]	144	96	99	15
Iwai (1999) [30]	167	96	99	NR
Longuet (2001) [26]	94	75	100	0
Wong (2002) [28]	19	100	90	12
Ishida (2003) [31]	152	93	99	0
Hwang (2003) [23]	36	100	100	0
Seethala (2005) [25]	61	74	100	7
Badoual (2006) [32]	694	79	99	12
Zbaeren (2008) [2]	110	93	95	0
Upton (2007) [33]	155	96	99	0

## Data Availability

The data presented in this study are available on request from the corresponding author.

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
