# Peer review of "Frozen Section of Parotid Gland Tumours: The Head and Neck Pathologist as a Key Member of the Surgical Team"

_jcm, 2022, doi:10.3390/jcm11051249_

Round 1
Reviewer 1 Report
Thank you very much for giving me the opportunity to review this manuscript on "Frozen section of parotid gland tumours: the head and neck pathologist as a key member of the surgical team". The study is on a clinically relevant topic and conducted on a large cohort coming from an experienced center and team.
My comments are:
Please explain, how an experienced pathologist was definded. Is there a special certification or how did you define a pathologist as experienced?
Can you provide a flow-chart of your pre- and intraoperative diagostic measures? Do you perform FNA or CNB in every patient? Do you perform FS routinely in every parotidectomy? Can you comment based on your experience on your suggestions for other hospitals performing parotidectomies?
Some minor spell checking is required (introduction: paroti -> parotid, spelling of tumour vs tumor -> use one way uniformly throughout the manuscript).
Author Response
Thank you very much for giving me the opportunity to review this manuscript on "Frozen section of parotid gland tumours: the head and neck pathologist as a key member of the surgical team". The study is on a clinically relevant topic and conducted on a large cohort coming from an experienced center and team.
My comments are:
Please explain, how an experienced pathologist was defined. Is there a special certification or how did you define a pathologist as experienced?
-We would like to thank you for this comment. In our context, experience in salivary gland pathology has a clinical and an academic aspect. In the last 14 years in our hospital, a certain pathologist receives the vast majority of all the salivary gland specimens for intraoperative pathologic consultation and verification of the diagnosis on definitive histology. This pathologist is an internationally known head and neck pathologist who performs 10-15 frozen sections of salivary gland pathologies per month and has more than 150 peer reviewed articles dealing with this demanding topic (h-index:49). This information has been added to the Materials and Methods.
Can you provide a flow-chart of your pre- and intraoperative diagostic measures? Do you perform FNA or CNB in every patient? Do you perform FS routinely in every parotidectomy? Can you comment based on your experience on your suggestions for other hospitals performing parotidectomies?
-We would like to thank you for this comment. In the event of preoperative suspicion of parotid gland malignancy or in cases with older, multimorbid patients and suspicion for a cystadenolymphoma (Warthin’s tumour), ultrasound-guided core needle biopsy of the lesion is performed in our department. Due to its low sensitivity, FNA has been abandoned as a diagnostic tool by parotid gland tumours in our department at all. In the case of confirmation of malignancy, an interdisciplinary tumor board takes place to define the best possible treatment. In cases of preoperatively known primary malignant tumors of the parotid gland (on core needle biopsy), a complete parotidectomy with neck dissection is performed. This surgical concept allows complete removal of the primary tumor and all possible locoregional metastatic foci to intraparotid and cervical lymph nodes. The same treatment modality is chosen in cases in which primary parotid gland malignancy is detected on intraoperative frozen section. In our department, frozen section is performed in cases with intraoperative suspicion for malignancy and aims at answering a series of questions (benign or malignant tumour, lymphoma or primary parotid carcinoma, low-grade or high-grade, nerve infiltration, resections margins – free of tumor). This information has been added to the Discussion. We hope that this explanation addresses your concerns.
Some minor spell checking is required (introduction: paroti -> parotid, spelling of tumour vs tumor -> use one way uniformly throughout the manuscript).
-Thank you for this comment. According to that, we replaced the word „tumor“ with the word „tumour“ and performed a spell checking throughout the manuscript.
We would like to thank you most kindly for your review and hope that this revision addresses your concerns.
Reviewer 2 Report
Thank you for the opportunity to read this work. This is a nice pathology manuscript, easily followed by the reader and very educational-the kind of manuscripts that pathologists appreciate. The information provided is indeed important for the routine diagnosis of salivary gland tumors.
I would have some suggestions:
1. The authors should provide the reason of discrepancy in the 16 cases (diagnostic or sampling error)
2. For a surgical pathology manuscript, pathology images are strikingly lacking. The authors should add represantitive FS images for better understanding. Even if the original FS are not available, there always is a block that could be cut.
3. It would be useful to add a Table with the studies existing on the same subject (how many cases, results, cause of misdiagnoses..)
4. In the discussion, for the Ref 17, the authors should state 25000 cases of all kind of tissues (and not salivary gland tumors only).
5. There is also an information missing: could the authors inform us either from the litterature or from their own archive, how many cases of parotidectomies require FS? For example, in their study there were almost 700 cases with frozen section/how many cases in total for the same period?
Author Response
Thank you for the opportunity to read this work. This is a nice pathology manuscript, easily followed by the reader and very educational-the kind of manuscripts that pathologists appreciate. The information provided is indeed important for the routine diagnosis of salivary gland tumors.
I would have some suggestions:
- The authors should provide the reason of discrepancy in the 16 cases (diagnostic or sampling error).
- We would like to thank you for this remark. According to that, we reviewed these 16 cases and added this information in the Table.
- For a surgical pathology manuscript, pathology images are strikingly lacking. The authors should add represantitive FS images for better understanding. Even if the original FS are not available, there always is a block that could be cut.
-Thank you for this comment. We added pathology images enlightening the problematic of frozen section by salivary gland tumors and hope that this addition addresses your concerns.
- It would be useful to add a Table with the studies existing on the same subject (how many cases, results, cause of misdiagnoses..)
-We would like to thank you for this remark. According to that, we added a Table with information on the relevant literature dealing with this subject. We hope that this addition addresses your concerns.
- In the discussion, for the Ref 17, the authors should state 25000 cases of all kind of tissues (and not salivary gland tumors only).
-Thank you for this remark. According to that, we added this information in the Discussion.
- There is also an information missing: could the authors inform us either from the litterature or from their own archive, how many cases of parotidectomies require FS? For example, in their study there were almost 700 cases with frozen section/how many cases in total for the same period?
We would like to thank you for this comment. We performed 669 frozen sections by a total of 2050 surgeries for parotid gland tumours in the same period of time.
We hope that our answers as well as the revision of the manuscript address your concerns.